# Maxing and Ranking with Few Assumptions

**Moein Falahatgar   Yi Hao   Alon Orlitsky   Venkatadheeraj Pichapati   Vaishakh Ravindrakumar**
University of California, San Deigo
{moein,yih179,alon,dheerajpv7,vaishakhr}@ucsd.edu

## Abstract

PAC maximum selection (maxing) and ranking of $n$ elements via random pairwise comparisons have diverse applications and have been studied under many models and assumptions. With just one simple natural assumption: strong stochastic transitivity, we show that maxing can be performed with linearly many comparisons yet ranking requires quadratically many. With no assumptions at all, we show that for the Borda-score metric, maximum selection can be performed with linearly many comparisons and ranking can be performed with $\mathcal{O}(n \log n)$ comparisons.

## 1 Introduction

### 1.1 Motivation

Maximum selection (maxing) and sorting using pairwise comparisons are among the most practical and fundamental algorithmic problems in computer science. As is well-known, maxing requires $n - 1$ comparisons, while sorting takes $\Theta(n \log n)$ comparisons.

The probabilistic version of this problem, where comparison outcomes are random, is of significant theoretical interest as well, and it too arises in many applications and diverse disciplines. In sports, pairwise games with random outcomes are used to determine the best, or the order, of teams or players. Similarly *Trueskill* [1] matches video gamers to create their ranking. It is also used for a variety of online applications such as to learn consumer preferences with the popular *A/B tests*, in recommender systems [2], for ranking documents from user clickthrough data [3, 4], and more. The popular crowd sourcing website GIFGIF [5] shows how pairwise comparisons can help associate emotions with many animated GIF images. Visitors are presented with two images and asked to select the one that better corresponds to a given emotion. For these reasons, and because of its intrinsic theoretical interest, the problem received a fair amount of attention.

### 1.2 Terminology and previous results

One of the first studies in the area, [6] assumed $n$ totally-ordered elements, where each comparison errs with the same, known, probability $\alpha < \frac{1}{2}$. It presented a maxing algorithm that uses $\mathcal{O}(\frac{n}{\alpha^2} \log \frac{1}{\delta})$ comparisons to output the maximum with probability $\geq 1 - \delta$, and a ranking algorithm that uses $\mathcal{O}(\frac{n}{\alpha^2} \log \frac{n}{\delta})$ comparisons to output the ranking with probability $\geq 1 - \delta$.

These results have been and continue to be of great interest. Yet this model has two shortcomings. It assumes that there is only one random comparison probability, $\alpha$, and that its value is known. In practice, comparisons have different, and arbitrary, probabilities, and they are not known in advance. To address more realistic scenarios, researchers considered more general probabilistic models.

Consider a set of $n$ elements, without loss of generality $[n] \stackrel{\text{def}}{=} \{1, 2, \ldots, n\}$. A *probabilistic model*, or *model* for short, is an assignment of a *preference probability* $p_{i,j} \in [0, 1]$ for every $i \neq j \in [n]$, reflecting the probability that $i$ is *preferred* when compared with $j$. We assume that repeated comparisons are independent and that there are no "draws", hence $p_{j,i} = 1 - p_{i,j}$.

If $p_{i,j} \geq \frac{1}{2}$, we say that $i$ is *preferable* to $j$ and write $i \geq j$. Element $i$ is *maximal* in a model if $i \geq j$ for all $j \neq i$. And a permutation $\ell_1, \ldots, \ell_n$ is a *ranking* if $\ell_i \geq \ell_j$ for all $i \leq j$. Observe that the first element of any ranking is always maximal. For example, for $n = 3$, $p_{1,2} = 1/2$, $p_{1,3} = 1/3$, and $p_{2,3} = 2/3$, we have $1 \geq 2$, $2 \geq 1$, $3 \geq 1$, and $2 \geq 3$. Hence 2 is the unique maximum, and 2,3,1 is the unique ranking. We seek algorithms that without knowing the underlying model, use pairwise comparisons to find a maximal element and a ranking.

Two concerns spring to mind. First, there may be two elements $i, j$ with $p_{i,j}$ arbitrarily close to half, requiring arbitrarily many comparisons just to determine which is preferable to the other. This concern has a common remedy, that we also adopt. The *PAC* paradigm, e.g. [7, 8], that requires the algorithm's output to be only *Probably Approximately Correct*.

Let $\tilde{p}_{i,j} \overset{\text{def}}{=} p_{i,j} - \frac{1}{2}$ be the *centered* preference probability. Note that $\tilde{p}_{i,j} \geq 0$ iff $i$ is preferable to $j$. If $\tilde{p}_{i,j} \geq -\epsilon$ we say that $i$ is $\epsilon$-*preferable* to $j$. For $0 < \epsilon < 1/2$, an element $i \in [n]$ is $\epsilon$-*maximum* if it is $\epsilon$-preferable to all other elements, namely, $\tilde{p}_{i,j} \geq -\epsilon \ \forall j \neq i$. Given $\epsilon > 0, \frac{1}{2} \geq \delta > 0$, a PAC maxing algorithm must output an $\epsilon$-maxima with probability $\geq 1 - \delta$, henceforth abbreviated *with high probability (WHP)*. Similarly, a permutation $\ell_1, \ldots, \ell_n$ of $\{1, \ldots, n\}$ is an $\epsilon$-*ranking* if $\ell_i$ is $\epsilon$-preferable to $\ell_j$ for all $i \leq j$, and a PAC ranking algorithm must output an $\epsilon$-ranking WHP. Note that in this paper, we consider $\delta \leq \frac{1}{2}$, the more practical regime. For larger values of $\delta$, one can use our algorithms with $\delta = \frac{1}{2}$.

The second concern is that not all models have a ranking, or even a maximal element. For example, for $p_{1,2} = p_{2,3} = p_{3,1} = 1$, or the more opaque yet interesting non-transitive coins [9], each element is preferable to the cyclically next, hence there is no maximal element and no ranking.

A standard approach, that again we too will adopt, to address this concern is to consider structured models. The simplest may be parametric models, of which one of the more common is *Placket Luce* (PL) [10, 11], where each element $i$ is associated with an unknown positive number $a_i$ and $p_{i,j} = \frac{a_i}{a_i + a_j}$. [12] derived a PAC maxing algorithm that uses $\mathcal{O}(\frac{n}{\epsilon^2} \log \frac{n}{\epsilon \delta})$ comparisons and a PAC ranking algorithm that uses $\mathcal{O}(\frac{n}{\epsilon^2} \log n \log \frac{n}{\epsilon \delta})$ comparisons for any PL model. Related results for the *Mallows model* under a non-PAC paradigm were derived by [13].

But significantly more general, and more realistic, non-parametric, models may also have maxima and rankings. A model is *strongly stochastically transitive (SST)*, if $i \geq j$ and $j \geq k$ imply $p_{i,k} \geq \max(p_{i,j}, p_{j,k})$. By simple induction, every SST model has a maximum element and a ranking. And one additional property, that is perhaps more difficult to justify, has proved helpful in constructing maxing and sorting PAC algorithms. A tournament satisfies the *stochastic triangle inequality* if $i \geq j$ and $j \geq k$ imply that $\tilde{p}_{i,k} \leq \tilde{p}_{i,j} + \tilde{p}_{j,k}$.

In Section 4 we show that if a model has a ranking, then an $\epsilon$-ranking can be found WHP via $\mathcal{O}(\frac{n^2}{\epsilon^2} \log \frac{n}{\delta})$ comparisons. For all models that satisfy both SST and triangle inequality, [7] derived a PAC maxing algorithm that uses $\mathcal{O}(\frac{n}{\epsilon^2} \log \frac{n}{\epsilon \delta})$ comparisons. [14] eliminated the $\log \frac{n}{\epsilon}$ factor and showed that $\mathcal{O}(\frac{n}{\epsilon^2} \log \frac{1}{\delta})$ comparisons suffice and are optimal, and constructed a nearly-optimal PAC ranking algorithm that uses $\mathcal{O}(\frac{n \log n (\log \log n)^3}{\epsilon^2})$ comparisons for all $\delta \geq \frac{1}{n}$, off by a factor of $\mathcal{O}((\log \log n)^3)$ from optimum. Lower-bounds follow from an analogy to [15, 6]. Observe that since the PL model satisfies both SST and triangle inequality, these results also improve the corresponding PL results.

Finally, we consider models that are not SST, or perhaps don't have maximal elements, rankings, or even their $\epsilon$-equivalents. In all these cases, one can apply a weaker order relation. The *Borda score* $s(i) \overset{\text{def}}{=} \frac{1}{n} \sum_j p_{i,j}$ is the probability that $i$ is preferable to another, randomly selected, element. Element $i$ is *Borda maximal* if $s(i) = \max_j s(j)$, and $\epsilon$-*Borda maximal* if $s(i) \geq \max_j s(j) - \epsilon$. A PAC Borda-maxing algorithm outputs an $\epsilon$-Borda maximal element WHP (with probability $\geq 1 - \delta$). Similarly, a *Borda ranking* is a permutation $i_1, \ldots, i_n$ such that for all $1 \leq j \leq n-1$, $s(i_j) \geq s(i_{j+1})$. An $\epsilon$-*Borda ranking* is a permutation where for all $1 \leq j \leq k \leq n$, $s(i_j) \geq s(i_k) - \epsilon$. A PAC Borda-ranking algorithm outputs an $\epsilon$-Borda ranking WHP.

Recall that Borda scores apply to all models. As noted in [16, 17, 8, 18] considering elements with nearly identical Borda scores shows that exact Borda-maxing and ranking requires arbitrarily many comparisons. [8] derived a PAC Borda ranking, and therefore also maxing, algorithms that use

$\mathcal{O}(\frac{n^2}{\epsilon^2})$ comparisons. [19] derived a $\mathcal{O}(\frac{n\log n}{\epsilon^2}\log(\frac{n}{\delta}))$ PAC Borda ranking algorithm for restricted setting. However note that several simple models, including $p_{1,2} = p_{2,3} = p_{3,1} = 1$ do not belong to this model.

[20, 21, 22] considered deterministic adversarial versions of this problem that has applications in [23]. Finally, we note that all our algorithms are adaptive, where each comparison is chosen based on the outcome of previous comparisons. Non-adaptive algorithms were discussed in [24, 25, 26, 27].

## 2   Results and Outline

Our goal is to find the minimal assumptions that enable efficient algorithms for these problems. In particular, we would like to see if we can eliminate the somewhat less-natural triangle inequality. With two algorithmic problems: maxing and ranking, and one property–SST and one special metric– Borda scores, the puzzle consists of four main questions.

1) With just SST (and no triangle inequality) are there: a) PAC maxing algorithms with $\mathcal{O}(n)$ comparisons? b) PAC ranking algorithms with near $\mathcal{O}(n\log n)$ comparisons? 2) With no assumptions at all, but for the Borda-score metric, are there: a) PAC Borda-maxing algorithms with $\mathcal{O}(n)$ comparisons? b) PAC Borda-ranking algorithms with near $\mathcal{O}(n\log n)$ comparisons?

We essentially resolve all four questions. 1a) Yes. In Section 3, Theorem 6, we use SST alone to derive a $\mathcal{O}\left(\frac{n}{\epsilon^2}\log\frac{1}{\delta}\right)$ comparisons PAC maxing algorithm. Note that this is the same complexity as with triangle inequality, and it matches the lower bound. 1b) No. In Section 4, Theorem 7, we show that there are SST models where any PAC ranking algorithm with $\epsilon \le 1/4$ requires $\Omega(n^2)$ comparisons. This is significantly higher than the roughly $\mathcal{O}(n\log n)$ comparisons needed with triangle inequality, and is close to the $\mathcal{O}(n^2\log n)$ comparisons required without any assumptions. 2a) Yes. In Section 5, Theorem 8, we derive a PAC Borda maxing algorithm that without any model assumptions requires $\mathcal{O}\left(\frac{n}{\epsilon^2}\log\frac{1}{\delta}\right)$ comparisons which is order optimal. 2b) Yes. In Section 5, Theorem 9, we derive a PAC Borda ranking algorithm that without any model assumptions requires $\mathcal{O}\left(\frac{n}{\epsilon^2}\log\frac{n}{\delta}\right)$ comparisons.

Beyond the theoretical results sections, in Section 6, we provide experiments on simulated data. In Section 7, we discuss the results.

## 3   Maxing

### 3.1   SEQ-ELIMINATE

Our main building block is a simple, though sub-optimal, algorithm SEQ-ELIMINATE that sequentially eliminates one element from input set to find an $\epsilon$-maximum under SST.

SEQ-ELIMINATE uses $\mathcal{O}\left(\frac{n}{\epsilon^2}\log\frac{n}{\delta}\right)$ comparisons and w.p.$\ge 1-\delta$, finds an $\epsilon$-maximum. Even for simpler models [15] we know that an algorithm needs $\Omega\left(\frac{n}{\epsilon^2}\log\frac{1}{\delta}\right)$ comparisons to find an $\epsilon$-maximum w.p.$\ge 1-\delta$. Hence the number of comparisons used by SEQ-ELIMINATE is optimal up to a constant factor when $\delta \le \frac{1}{n}$ but can be $\log n$ times the lower bound for $\delta = \frac{1}{2}$.

By SST, any element that is $\epsilon$-preferable to absolute maximum element of $S$ is an $\epsilon$-maximum of $S$. Therefore if we can reduce $S$ to a subset $S'$ of size $\mathcal{O}\left(\frac{n}{\log n}\right)$ that contains an absolute maximum of $S$ using $\mathcal{O}\left(\frac{n}{\epsilon^2}\log\frac{1}{\delta}\right)$ comparisons, we can then use SEQ-ELIMINATE to find an $\epsilon$-maximum of $S'$ and the number of comparisons is optimal up to constants. We provide one such reduction in subsection 3.2.

Sequential elimination techniques have been used before [13] to find an absolute maximum. In such approaches, a running element is maintained, and is compared and replaced with a competing element in $S$ if the latter is found to be better with confidence $\ge 1 - \delta/n$. Note that if the running and competing elements are close to each other, this technique can take an arbitrarily long time to declare the winner. But since we are interested in finding only an $\epsilon$-maximum, SEQ-ELIMINATE circumvents this issue. We later show that SEQ-ELIMINATE needs to update the running element $r$ with the competing element $c$ if $\tilde{p}_{c,r} \ge \epsilon$ and retain $r$ if $\tilde{p}_{c,r} \le 0$. If $0 < \tilde{p}_{c,r} < \epsilon$, replacing or

retaining $r$ doesn't affect the performance of SEQ-ELIMINATE significantly. Thus, in other words we've reduced the problem to testing whether $\tilde{p}_{c,r} \leq 0$ or $\tilde{p}_{c,r} \geq \epsilon$.

Assuming that testing problem always returns the right answer, since SEQ-ELIMINATE never replaces the running element with a worse element, either the output is the absolute maximum $b^*$ or $b^*$ is never the running element. If $b^*$ is eliminated against running element $r$ then $\tilde{p}_{b^*,r} \leq \epsilon$ and hence $r$ is an $\epsilon$-maximum and since the running element only gets better, the output is an $\epsilon$-maximum.

We first present a testing procedure COMPARE that we use to update the running element in SEQ-ELIMINATE.

### 3.1.1 COMPARE

COMPARE$(i,j,\epsilon_l,\epsilon_u,\delta)$ takes two elements $i$ and $j$, and two biases $\epsilon_u > \epsilon_l$, and with confidence $\geq 1 - \delta$, determines whether $\tilde{p}_{i,j}$ is $\leq \epsilon_l$ or $\geq \epsilon_u$.

For this, COMPARE compares the two elements $2/(\epsilon_u - \epsilon_l)^2 \log(2/\delta)$ times. Let $\hat{p}_{i,j}$ be the fraction of times $i$ beats $j$, and let $\hat{\tilde{p}}_{i,j} \stackrel{\text{def}}{=} \hat{p}_{i,j} - \frac{1}{2}$. If $\hat{\tilde{p}}_{i,j} < (\epsilon_l + \epsilon_u)/2$, COMPARE declares $\tilde{p}_{i,j} \leq \epsilon_l$ (returns 1), and otherwise it declares $\tilde{p}_{i,j} \geq \epsilon_u$ (returns 2).

Due to lack of space, we present the algorithm COMPARE in Appendix A.1 along with certain improvements for better performance in practice .

In the Lemma below, we bound the number of comparisons used by COMPARE and prove its correctness. Proof is in A.2.

**Lemma 1.** *For $\epsilon_u > \epsilon_l$, COMPARE$(i,j,\epsilon_l,\epsilon_u,\delta)$ uses $\leq \frac{2}{(\epsilon_u - \epsilon_l)^2} \log \frac{2}{\delta}$ comparisons and if $\tilde{p}_{i,j} \leq \epsilon_l$, then w.p.$\geq 1 - \delta$, it returns 1, else if $\tilde{p}_{i,j} \geq \epsilon_u$, w.p.$\geq 1 - \delta$, it returns 2.*

Now we present SEQ-ELIMINATE that uses the testing subroutine COMPARE and finds an $\epsilon$-maximum.

### 3.1.2 SEQ-ELIMINATE Algorithm

SEQ-ELIMINATE takes a variable set $S$, selects a random *running element* $r \in S$ and repeatedly uses COMPARE$(c,r,0,\epsilon,\delta/n)$ to compare $r$ to a random *competing* element $c \in S \setminus r$. If COMPARE returns 1 i.e., deems $\tilde{p}_{c,r} \leq 0$, it retains $r$ as the running element and eliminates $c$ from $S$, but if COMPARE returns 2 i.e., deems $\tilde{p}_{c,r} \geq \epsilon$, it eliminates $r$ from $S$ and updates $c$ as the new running element.

---

**Algorithm 1** SEQ-ELIMINATE

1: **inputs**
2:     Set $S$, bias $\epsilon$, confidence $\delta$
3: $n \leftarrow |S|$
4: $r \leftarrow$ a random $c \in S$, $S = S \setminus \{r\}$
5: **while** $S \neq \varnothing$ **do**
6:     Pick a random $c \in S$, $S = S \setminus \{c\}$.
7:     **if** COMPARE$(c,r,0,\epsilon,\frac{\delta}{n}) = 2$ **then**
8:         $r \leftarrow c$
9:     **end if**
10: **end while**
11: **return** r

---

We now bound the number of comparisons used by SEQ-ELIMINATE$(S,\epsilon,\delta)$ and prove its correctness. Proof is in A.3.

**Theorem 2.** *SEQ-ELIMINATE$(S,\epsilon,\delta)$ uses $\mathcal{O}\left(\frac{|S|}{\epsilon^2} \log \frac{|S|}{\delta}\right)$ comparisons, and w.p.$\geq 1 - \delta$ outputs an $\epsilon$-maximum.*

## 3.2 Reduction

Recall that, for $\delta \leq \frac{1}{n}$, SEQ-ELIMINATE is order-wise optimal. For $\delta \geq \frac{1}{n}$, here we present a reduction procedure that uses $\mathcal{O}\left(\frac{n}{\epsilon^2}\log\frac{1}{\delta}\right)$ comparisons and w.p.$\geq 1 - \delta$, outputs a subset $S'$ of size $\mathcal{O}(\sqrt{n \log n})$ and an element $a$ such that either $a$ is a $2\epsilon/3$-maximum or $S'$ contains an absolute maximum of $S$. Combining the reduction with SEQ-ELIMINATE results in an order-wise optimal algorithm.

We form the reduced subset $S'$ by pruning $S$. We compare each element $e \in S$ with an anchor element $a$, test whether $\tilde{p}_{e,a} \leq 0$ or $\tilde{p}_{e,a} \geq 2\epsilon/3$ using COMPARE, and retain all elements $e$ for which COMPARE returns the second hypothesis. For $S'$ to be of size $\mathcal{O}(\sqrt{n \log n})$ we'd like to pick an anchor element that is among the top $\mathcal{O}(\sqrt{n \log n})$ elements. But this can be computationally hard and we show that it suffices to pick an anchor that is not $\epsilon/3$-preferable to at most $\mathcal{O}(\sqrt{n \log n})$ elements in $S$.

An element $a$ is called an $(\epsilon, n')$-*good anchor* if $a$ is not $\epsilon$-preferable to at most $n'$ elements, i.e., $|\{e : e \in S \text{ and } \tilde{p}_{e,a} > \epsilon\}| \leq n'$.

We now present the subroutine PICK-ANCHOR that finds a good anchor element.

### 3.2.1 Picking Anchor Element

PICK-ANCHOR$(S, n', \epsilon, \delta)$ uses $\mathcal{O}\left(\frac{n}{n'\epsilon^2}\log\frac{1}{\delta}\log\frac{n}{n'\delta}\right)$ comparisons and w.p.$\geq 1 - \delta$, outputs an $(\epsilon, n')$-good anchor element. PICK-ANCHOR first picks randomly a set $Q$ of $\frac{n}{n'}\log\frac{2}{\delta}$ elements from $S$ without replacement. This ensures that w.p.$\geq 1 - \delta$, $Q$ contains at least one of the top $n'$ elements. We then use SEQ-ELIMINATE to find an $\epsilon$-maximum of $Q$.

Let the absolute maximum element of $Q$ be denoted as $q^*$. Now an $\epsilon$-maximum of $Q$ is $\epsilon$-preferable to $q^*$. Further, if $Q$ contains an element in the top $n'$ elements, there exists $n - n'$ elements worse than $q^*$ in $S$. Thus by SST, the $\epsilon$-maximum of $Q$ is also $\epsilon$-preferable to these $n - n'$ elements and hence the output of PICK-ANCHOR is an $(\epsilon, n')$-good anchor element. PICK-ANCHOR is shown in appendix A.4

We now bound the number of comparisons used by PICK-ANCHOR and prove its correctness. Proof is in A.5.

**Lemma 3.** PICK-ANCHOR$(S, n', \epsilon, \delta)$ *uses* $\mathcal{O}\left(\frac{n}{n'\epsilon^2}\log\frac{1}{\delta}\log\frac{n}{n'\delta}\right)$ *comparisons and w.p.*$\geq 1 - \delta$, *outputs an* $(\epsilon, n')$-*good anchor element.*

**Remark 4.** *Note that* PICK-ANCHOR$(S, cn, \epsilon, \delta)$ *uses* $\mathcal{O}_c\left(\frac{1}{\epsilon^2}\left(\log\frac{1}{\delta}\right)^2\right)$ *comparisons where the constant depends only on $c$ but not on $n$. Hence it is advantageous to use this method to pick near-maximum element when $n$ is large.*

We now present PRUNE that takes an anchor element as input and prunes the set $S$ using the anchor.

### 3.2.2 Pruning

Given an $(\epsilon_l, n')$-good anchor element $a$, w.p.$\geq 1 - \delta/2$, PRUNE$(S, a, n', \epsilon_l, \epsilon_u, \delta)$ outputs a subset $S'$ of size $\leq 2n'$. Further, any element $e$ that is at least $\epsilon_u$-better than $a$ i.e., $\tilde{p}_{e,a} \geq \epsilon_u$ is in $S'$ w.p.$\geq 1 - \delta/2$.

PRUNE prunes $S$ in multiple rounds. In each round $t$, for every element $e$ in $S$, PRUNE tests whether $\tilde{p}_{e,a} \leq \epsilon_l$ or $\tilde{p}_{e,a} \geq \epsilon_u$ using COMPARE$(e, a, \epsilon_l, \epsilon_u, \delta/2^{t+1})$ and eliminates $e$ if the first hypothesis i.e., $\tilde{p}_{e,a} \leq \epsilon_l$ is returned. By Lemma 1, an element $e$ that is $\epsilon_u$ better than $a$ i.e., $\tilde{p}_{e,a} \geq \epsilon_u$ passes the $t^{th}$ round of pruning w.p.$\geq 1 - \delta/2^{t+1}$. Thus by union bound, the probability that such an element is not present in the pruned set is $\leq \sum_{t=1}^{\infty} \delta/2^{t+1} \leq \delta/2$.

Now for element $e$ that is not $\epsilon_l$-better than $a$ i.e., $\tilde{p}_{e,a} \leq \epsilon_l$, by Lemma 1, the first hypothesis is returned w.p.$\geq 1 - \delta/4$. Hence w.h.p., the number of such elements (not $\epsilon_l$-better elements) is reduced by a factor of $\delta$ after each round. Since $a$ is an $(\epsilon_l, n')$-good anchor element, there are at most $n'$ elements atleast $\epsilon_l$-better than $a$. Thus the number of elements left in the pruned set after round $t$ is at most $n' + n\delta^t$. Thus PRUNE succeeds eventually in reducing the size to $\leq 2n'$ (in $\leq \log_{1/\delta}\frac{n}{n'}$ rounds).

---

**Algorithm 2** PRUNE

1: **inputs**
2:     Set $S$, element $a$, size $n'$, lower bias $\epsilon_l$, upper bias $\epsilon_u$, confidence $\delta$.
3: $t \leftarrow 1$
4: $S_1 \leftarrow S$
5: **while** $|S_t| > 2n'$ and $t < \log^2 n$ **do**
6:     **Initialize:** $Q_t \leftarrow \varnothing$
7:     **for** $e$ in $S_t$ **do**
8:         **if** COMPARE$(e, a, \epsilon_l, \epsilon_u, \delta/2^{t+1}) = 1$ **then**
9:             $Q_t \leftarrow Q_t \bigcup \{e\}$
10:        **end if**
11:     **end for**
12:     $S_{t+1} \leftarrow S_t \smallsetminus Q_t$
13:     $t \leftarrow t + 1$
14: **end while**
15: **return** $S_t$.

---

We now bound the number of comparisons used by PRUNE and prove its correctness. Proof is in A.6.

**Lemma 5.** *If $n' \geq \sqrt{6n \log n}$, $\delta \geq \frac{1}{n}$ and $a$ is an $(\epsilon_l, n')$-good anchor element, then w.p.$\geq 1 - \frac{\delta}{2}$, PRUNE$(S, a, n', \epsilon_l, \epsilon_u, \delta)$ uses $\mathcal{O}\big(\frac{n}{(\epsilon_u - \epsilon_l)^2} \log \frac{1}{\delta}\big)$ comparisons and outputs a set of size less than $2n'$. Further if $a$ is not an $\epsilon_u$-maximum of $S$ then w.p.$\geq 1 - \frac{\delta}{2}$, the output set contains an absolute maximum element of $S$.*

### 3.3 Full Algorithm

We now present the main algorithm, OPT-MAXIMIZE that w.p.$\geq 1 - \delta$, uses $\mathcal{O}\big(\frac{n}{\epsilon^2} \log \frac{1}{\delta}\big)$ comparisons and outputs an $\epsilon$-maximum. For $\delta \leq \frac{1}{n}$, SEQ-ELIMINATE uses $\mathcal{O}(\frac{n}{\epsilon^2} \log \frac{1}{\delta})$ comparisons and hence we directly use SEQ-ELIMINATE. Below we assume $\delta > \frac{1}{n}$.

Here OPT-MAXIMIZE first finds an $(\epsilon/3, \sqrt{6n \log n})$-good anchor element $a$ using PICK-ANCHOR$(S, \sqrt{6n \log n}, \epsilon/3, \frac{\delta}{4})$. Then using PRUNE$(S, a, \sqrt{6n \log n}, \epsilon/3, 2\epsilon/3, \frac{\delta}{4})$ with $a$, OPT-MAXIMIZE prunes $S$ to a subset $S'$ of size $\leq 2\sqrt{6n \log n}$ such that if $a$ is not a $2\epsilon/3$ maximum i.e. $\tilde{p}_{b^*, a} > 2\epsilon/3$, $S'$ contains the absolute maximum $b^*$ w.p.$\geq 1 - \delta/2$. OPT-MAXIMIZE then checks if $a$ is a $2\epsilon/3$ maximum by using COMPARE$(e, a, 2\epsilon/3, \epsilon, \delta/(4n))$ for every element $e \in S'$. If COMPARE returns first hypothesis for every $e \in S'$ then OPT-MAXIMIZE outputs $a$ or else OPT-MAXIMIZE outputs SEQ-ELIMINATE$(S', \epsilon, \frac{\delta}{4})$.

Note that only one of these three cases is possible: (1) $a$ is a $2\epsilon/3$-maximum, (2) $a$ is not an $\epsilon$-maximum and (3) $a$ is an $\epsilon$-maximum but not a $2\epsilon/3$-maximum. In case (1), since $a$ is a $2\epsilon/3$-maximum, by Lemma 1, w.p.$\geq 1 - \delta/4$, COMPARE returns the first hypothesis for every $e \in S'$ and OPT-MAXIMIZE outputs $a$. In both cases (2) and (3), as stated above, w.p.$\geq 1 - \delta/2$, $S'$ contains the absolute maximum $b^*$. Now in case (2) since $a$ is not an $\epsilon$-maximum, by Lemma 1, w.p.$\geq 1 - \delta/(4n)$, COMPARE$(b^*, a, 2\epsilon/3, \epsilon, \delta/(4n))$ returns the second hypothesis. Thus OPT-MAXIMIZE outputs SEQ-ELIMINATE$(S', \epsilon, \delta/4)$, which w.p.$\geq 1 - \delta/4$, returns an $\epsilon$-maximum of $S'$ (recall that an $\epsilon$-maximum of $S'$ is an $\epsilon$-maximum of $S$ if $S'$ contains $b^*$). Finally in case (3), OPT-MAXIMIZE either outputs $a$ or SEQ-ELIMINATE$(S', \epsilon, \delta/4)$ and either output is an $\epsilon$-maximum w.p.$\geq 1 - \delta$. In the below Theorem, we bound comparisons used by OPT-MAXIMIZE and prove its correctness. Proof is in A.7.

**Theorem 6.** *W.p.$\geq 1 - \delta$, OPT-MAXIMIZE$(S, \epsilon, \delta)$ uses $\mathcal{O}(\frac{n}{\epsilon^2} \log \frac{1}{\delta})$ comparisons and outputs an $\epsilon$-maximum.*

## 4 Ranking

Recall that [14] considered a model with both SST and stochastic triangle inequality and derived an $\epsilon$-ranking with $\mathcal{O}\big(\frac{n \log n (\log \log n)^3}{\epsilon^2}\big)$ comparisons for $\delta = \frac{1}{n}$. By constrast, we consider a more

---

**Algorithm 3** OPT-MAXIMIZE

---
1: **inputs**
2:     Set $S$, bias $\epsilon$, confidence $\delta$.
3: **if** $\delta \leq \frac{1}{n}$ **then**
4:     **return** SEQ-ELIMINATE$(S, \epsilon, \delta)$
5: **end if**
6: $a \leftarrow$ PICK-ANCHOR$(S, \sqrt{6n \log n}, \epsilon/3, \frac{\delta}{4})$
7: $S' \leftarrow$ PRUNE$(S, a, \sqrt{6n \log n}, \epsilon/3, 2\epsilon/3, \frac{\delta}{4})$
8: **for** element $e$ in $S'$ **do**
9:     **if** COMPARE$(e, a, \frac{2\epsilon}{3}, \epsilon, \frac{\delta}{4n}) = 2$ **then**
10:         **return** SEQ-ELIMINATE$(S', \epsilon, \frac{\delta}{4})$
11:     **end if**
12: **end for**
13: **return** $a$

---

general model without stochastic triangle inequality and show that even a $1/4$-ranking with just SST takes $\Omega(n^2)$ comparisons for $\delta \leq \frac{1}{8}$.

To establish the lower bound, we reduce the problem of finding $1/4$-ranking to finding a coin with bias 1 among $\frac{n(n-1)}{2} - 1$ other fair coins. For this, we consider the following model with $n$ elements $\{a_1, a_2, ..., a_n\}$: $\tilde{p}_{a_1, a_n} = \frac{1}{2}$, $\tilde{p}_{a_i, a_j} = \mu (0 < \mu < 1/n^{10})$, when $i < j$ and $(i, j) \neq (1, n)$. Note that this model satisfies SST but not stochastic triangle inequality. Also note that any ranking where $a_1$ precedes $a_n$ is an $1/4$-ranking and thus the algorithm only needs to order $a_1$ and $a_n$ correctly. Now the output of a comparison between any two elements other than $a_1$ and $a_n$ is essentially a fair coin toss (since $\mu$ is very small). Thus if we output a ranking without querying comparison between $a_1$ and $a_n$, then the ranking is correct w.p.$\approx \frac{1}{2}$ since $a_1$ and $a_n$ must necessarily be ordered correctly. Now if an algorithm uses only $n^2/20$ comparisons then the probability that the algorithm queried at least one comparison between $a_1$ and $a_n$ is less than $\frac{1}{2}$ and hence cannot achieve a confidence of $\frac{7}{8}$. Proof sketch in B.1.

**Theorem 7.** *There exists a model that satisfies SST for which any algorithm requires $\Omega(n^2)$ comparisons to find a $1/4$-ranking with probability $\geq 7/8$.*

We also present a trivial $\epsilon$-ranking algorithm in Appendix B.2 that for any stochastic model with ranking (Weak Stochastic Transitivity), uses $\mathcal{O}(\frac{n^2}{\epsilon^2} \log \frac{n}{\delta})$ comparisons and outputs an $\epsilon$-ranking w.p.$\geq 1 - \delta$.

## 5 Borda Scores

We show that for general models, using $\mathcal{O}(\frac{n}{\epsilon^2} \log \frac{1}{\delta})$ comparisons w.p.$\geq 1 - \delta$, we can find an $\epsilon$-Borda maximum and using $\mathcal{O}(\frac{n}{\epsilon^2} \log \frac{n}{\delta})$ comparisons w.p.$\geq 1 - \delta$, we can find an $\epsilon$-Borda ranking.

Recall that Borda score $s(e)$ of an element $e$ is the probability that $e$ is preferable to an element picked randomly from $S$ i.e., $s(e) = \frac{1}{n} \sum_{f \in S} \tilde{p}_{e, f}$. We first make a connection between Borda scores of elements and the traditional multi armed bandit setting. In the Bernoulli multi armed setting, every arm $a$ is associated with a parameter $q(a)$ and pulling that arm results in a reward $B(q(a))$, a Bernoulli random variable with parameter $q(a)$. Observe that we can simulate our pairwise comparisons setting as a traditional bandit arms setting by comparing an element with a random element where in our setting, for every element $e$, the associated parameter is $s(e)$. Thus PAC optimal algorithms derived under traditional bandit setting work for PAC Borda score setting too. [28] and several others derived a PAC maximum arm selection algorithms that use $\mathcal{O}(\frac{n}{\epsilon^2} \log \frac{1}{\delta})$ comparisons and find an arm with parameter at most $\epsilon$ less than the highest. This implies an $\epsilon$-Borda maxing algorithm with the same complexity. Proof follows from reduction to Bernoulli multi-armed bandit setting.

**Theorem 8.** *There exists an algorithm that uses $\mathcal{O}(\frac{n}{\epsilon^2} \log \frac{1}{\delta})$ comparisons and w.p.$\geq 1 - \delta$, outputs an $\epsilon$-Borda maximum.*

For $\epsilon$-Borda ranking, we note that if we compare an element $e$ with $\frac{2}{\epsilon^2}\log\frac{2n}{\delta}$ random elements, w.p. $\geq 1 - \delta/n$, the fraction of times $e$ wins approximates the Borda score of $e$ to an additive error of $\frac{\epsilon}{2}$. Ranking based on these approximate scores results in an $\epsilon$-Borda ranking. We present BORDA-RANKING in C.1 that uses $\frac{2n}{\epsilon^2}\log\frac{2n}{\delta}$ comparisons and w.p.$\geq 1-\delta$ outputs an $\epsilon$-Borda ranking. Proof in C.1.

**Theorem 9.** BORDA-RANKING$(S,\epsilon,\delta)$ *uses* $\frac{2n}{\epsilon^2}\log\frac{2n}{\delta}$ *comparisons and w.p.$\geq 1-\delta$ outputs an $\epsilon$-Borda ranking.*

## 6 Experiments

In this section we validate the performance of our algorithms using simulated data. Since we essentially derived a negative result for $\epsilon$-ranking, we consider only our $\epsilon$-maxing algorithms - SEQ-ELIMINATE and OPT-MAXIMIZE for experiments. All results are averaged over 100 runs.

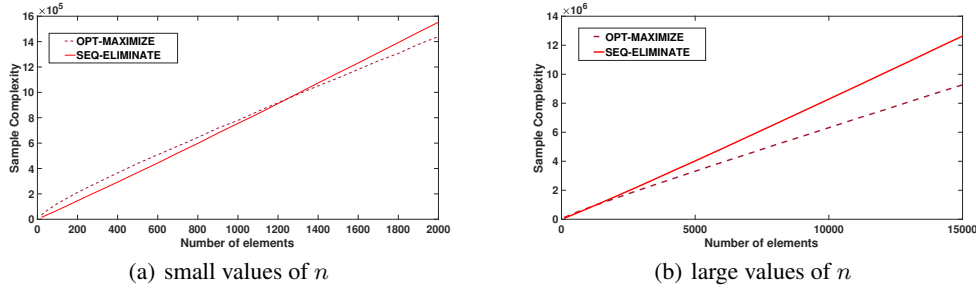

(a) small values of $n$  (b) large values of $n$

Figure 1: Comparison of SEQ-ELIMINATE and OPT-MAXIMIZE

Similar to [14, 7], we consider the stochastic model $p_{i,j} = 0.6 \ \forall i < j$. We use maxing algorithms to find 0.05-maximum with error probability $\delta = 0.1$. Note that $i = 1$ is the unique 0.05-maximum under this model. In Figure 1, we compare the performance of SEQ-ELIMINATE and OPT-MAXIMIZE over different ranges of $n$. Figures 1(a), 1(b) show that for small $n$ i.e., $n \leq 1300$ SEQ-ELIMINATE performs well and for large $n$ i.e., $n \geq 1300$, OPT-MAXIMIZE performs well. Since we are using $\delta = 0.1$, the experiment suggests that for $\delta \gtrsim \frac{1}{n^{1/3}}$, OPT-MAXIMIZE uses fewer comparisons as compared to SEQ-ELIMINATE. Hence it would be beneficial to use SEQ-ELIMINATE for $\delta \leq \frac{1}{n^{1/3}}$ and OPT-MAXIMIZE for higher values of $\delta$. In further experiments, we use $\delta = 0.1$ and $n < 1000$ so we use SEQ-ELIMINATE for better performance.

We compare SEQ-ELIMINATE with **BTM-PAC** [7], **KNOCKOUT** [14], **MallowsMPI** [13], and **AR** [16] . **KNOCKOUT** and **BTM-PAC** are PAC maxing algorithms for models with SST and stochastic triangle inequality requirements. **AR** finds an element with maximum Borda score. **Mallows** finds the absolute best element under Weak Stochastic Transitivity.

We again consider the model: $p_{i,j} = 0.6 \ \forall i < j$ and try to find a 0.05-maximum with error probability $\delta = 0.1$. Note that this model satisfies both SST and stochastic triangle inequality and under this model all these algorithms can find an $\epsilon$-maximum. From Figure 2(a), we can see that **BTM-PAC** performs worse for even small values of $n$ and from Figure 2(b), we can see that **AR** performs worse for higher values of $n$. One possible reason is that **BTM-PAC** is tailored for reducing regret in the bandit setting and in the case of **AR**, Borda scores of elements become approximately the same with increasing number of elements, leading to more comparisons. For this reason, we drop **BTM-PAC** and **AR** for further experiments.

We also tried **PLPAC** [12] but it fails to achieve required accuracy of $1-\delta$ since it is designed primarily for Plackett-Luce. For example, we considered the previous setting $p_{i,j} = 0.6 \ \forall i < j$ with $n = 100$ and tried to find a 0.09-maximum with $\delta = 0.1$. Even though **PLPAC** used almost same number of comparisons (57237) as SEQ-ELIMINATE (56683), PLPAC failed to find 0.09-maxima 20 out of 100 runs whereas SEQ-ELIMINATE found the maximum in all 100 runs.

In figure 3, we compare algorithms SEQ-ELIMINATE, **KNOCKOUT** [14] and **MallowsMPI** [13] for models that do not satisfy stochastic triangle inequality. In Figure 3(a), we consider the stochastic model $p_{1,j} = \frac{1}{2} + \tilde{q} \ \forall j \leq n/2$, $p_{1,j} = 1 \ \forall j > n/2$ and $p_{i,j} = \frac{1}{2} + \tilde{q} \ \forall 1 < i < j$ where $\tilde{q} \leq 0.05$ and we pick $n = 10$. Observe that this model satisfies SST but not stochastic triangle inequality. Here

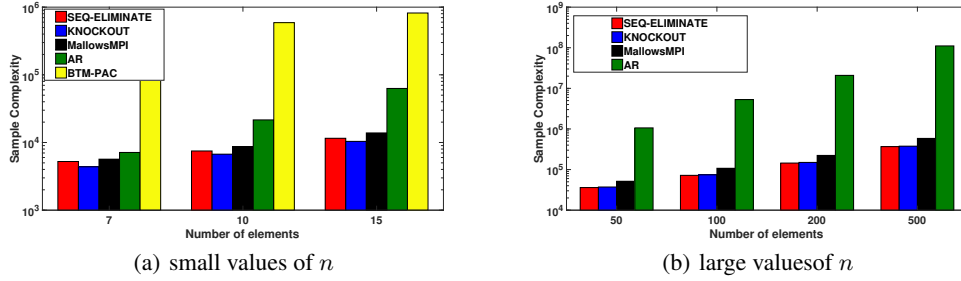

(a) small values of $n$                    (b) large valuesof $n$

Figure 2: Comparison of Maxing Algorithms with Stochastic Triangle Inequality

again, we try to find a 0.05-maximum with $\delta = 0.1$. Note that any $i \leq n/2$ is a 0.05 maximum. From Figure 3(a), we can see that **MallowsMPI** uses more comparisons as $\tilde{q}$ decreases since **MallowsMPI** is not a PAC algorithm and tries to find the absolute maximum. Even though **KNOCKOUT** performs better than **MallowsMPI**, it fails to output a 0.05 maximum with probability 0.12 for $\tilde{q} = 0.001$ and 0.26 for $\tilde{q} = 0.0001$. Thus **KNOCKOUT** can fail when the model doesn't satisfy stochastic triangle inequality. We give an explanation for this behavior in Appendix D. By constrast, even for $\tilde{q} = 0.0001$, SEQ-ELIMINATE outputted a 0.05 maximum in all runs and outputted the absoulte maximum in 76% of trials. We can also see that SEQ-ELIMINATE uses much fewer comparisons compared to the other two algorithms.

In Figure 3(b), we compare SEQ-ELIMINATE and **MallowsMPI** on the Mallows model, a model which doesn't satisfy stochastic triangle inequality. Mallows model can be specified with one parameter $\phi$. We consider $n = 10$ elements and find a 0.05-maximum with error probablility $\delta = 0.05$. From Figure 3(b) we can see that the performance of **MallowsMPI** gets worse as $\phi$ approaches 1, since comparison probabilities get close to $\frac{1}{2}$ whereas SEQ-ELIMINATE is not affected.

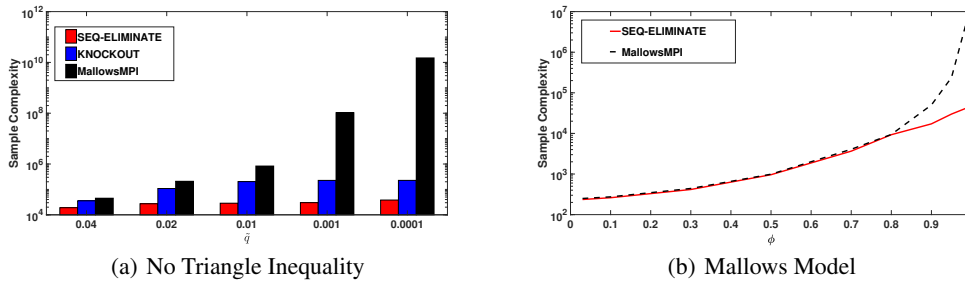

(a) No Triangle Inequality                    (b) Mallows Model

Figure 3: Comparison of SEQ-ELIMINATE and MALLOWSMPI over Mallows Model

One more experiment is presented in Appendix E.

## 7 Conclusion

We extended the study of PAC maxing and ranking to general models which satisfy SST but not stochastic triangle inequality. For PAC maxing, we derived an algorithm with linear complexity. For PAC ranking, we showed a negative result that any algorithm needs $\Omega(n^2)$ comparisons. We thus showed that removal of stochastic triangle inequality constraint does not affect PAC maxing but affects PAC ranking. We also ran experiments over simulated data and showed that our PAC maximum selection algorithms are better than other maximum selection algorithms.

For unconstrained models, we derived algorithms for PAC Borda maxing and PAC Borda ranking by making connections with traditional multi-armed bandit setting.

**Acknowledgments**

We thank NSF for supporting this work through grants CIF-1564355 and CIF-1619448.

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
