[Supplementary Material]

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

# A  Maxing

## A.1  COMPARE Algorithm

Motivated by a related algorithm in [14], we describe an adaptive version of COMPARE that stops when it is confident about the result, even if the number of comparisons is less than that specified in subsection 3.1.1. If $\tilde{p}_{i,j}$ is far outside $(\epsilon_l, \epsilon_u)$, this adaptive algorithm will terminate much sooner.

To do so, COMPARE maintains a varying confidence interval $\hat{c}$ such that w.p. $\geq 1 - \delta$, $|\hat{\tilde{p}}_{i,j} - \tilde{p}_{i,j}| < \hat{c}$ after any number of comparisons. If at any time before the above number of comparisons, $|\hat{\tilde{p}}_{i,j} - \frac{\epsilon_l + \epsilon_u}{2}| > \hat{c}$, COMPARE simply returns the result based on the current $\hat{\tilde{p}}_{i,j}$.

---

**Algorithm 4** COMPARE

1: **inputs**
2:    element $i$, element $j$, bias lower limit $\epsilon_l \geq 0$, bias upper limit $\epsilon_u > \epsilon_l$, confidence $\delta$
3: **initialize**
4:    $\epsilon_m = (\epsilon_l + \epsilon_u)/2$, $\hat{\tilde{p}}_{i,j} \leftarrow 0$, $\hat{c} \leftarrow \frac{1}{2}$, $t \leftarrow 0$, $w \leftarrow 0$
5: **while** $|\hat{\tilde{p}}_{i,j} - \epsilon_m| \leq \hat{c}$ and $t \leq \frac{2}{(\epsilon_u - \epsilon_l)^2} \log \frac{2}{\delta}$ **do**
6:    Compare $i$ and $j$
7:    **if** $i$ wins **then**
8:       $w \leftarrow w + 1$
9:    **end if**
10:    $t \leftarrow t + 1$
11:    $\hat{\tilde{p}}_{i,j} \leftarrow \frac{w}{t} - \frac{1}{2}$, $\hat{c} \leftarrow \sqrt{\frac{1}{2t} \log \frac{4t^2}{\delta}}$
12: **end while**
13: **if** $\hat{\tilde{p}}_{i,j} \leq \epsilon_m$ **then**
14:    **return** 1
15: **end if**
16: **return** 2

---

## A.2  Proof of Lemma 1

We prove Lemma by dividing it into smaller parts. We first bound the comparisons used by COMPARE.

**Lemma 10.** *For $\epsilon_u > \epsilon_l \geq 0$, COMPARE$(i, j, \epsilon_l, \epsilon_u, \delta)$ uses $\leq \frac{2}{(\epsilon_u - \epsilon_l)^2} \log \frac{2}{\delta}$ comparisons.*

*Proof.* Notice that COMPARE$(i, j, \epsilon_l, \epsilon_u, \delta)$ compares elements $i$ and $j$ for at most $m = \frac{2}{(\epsilon_u - \epsilon_l)^2} \log \frac{2}{\delta}$ times and hence the Lemma follows. □

We show that under the first hypothesis namely $\tilde{p}_{i,j} \leq \epsilon_l$, w.p.$\geq 1 - \delta$, COMPARE$(i, j, \epsilon_l, \epsilon_u, \delta)$ returns 1.

**Lemma 11.** *For $\epsilon_u > \epsilon_l \geq 0$, if $\tilde{p}_{i,j} \leq \epsilon_l$, then w.p.$\geq 1 - \delta$, COMPARE$(i, j, \epsilon_l, \epsilon_u, \delta)$ outputs 1.*

*Proof.* Let $\hat{p}_{i,j}^t$ and $\hat{c}^t$ denote $\hat{p}_{i,j}$ and $\hat{c}$ respectively after $t$ comparisons between $i$ and $j$ during COMPARE$(i, j, \epsilon_l, \epsilon_u, \delta)$. COMPARE$(i, j, \epsilon_l, \epsilon_u, \delta)$ outputs 2 only if $\hat{p}_{i,j}^t > \frac{1}{2} + \frac{\epsilon_l + \epsilon_u}{2} + \hat{c}^t$ for any $t < m = \frac{2}{(\epsilon_l - \epsilon_u)^2} \log \frac{2}{\delta}$ or if $\hat{p}_{i,j}^m > \frac{1}{2} + \frac{\epsilon_l + \epsilon_u}{2}$. We bound the probability of either of these events by $\frac{\delta}{2}$ and the result follows from the union bound.

By Hoeffding's inequality,

$$Pr\left(\hat{p}_{i,j}^t > \frac{1}{2} + \frac{\epsilon_l + \epsilon_u}{2} + \hat{c}^t\right) \leq Pr\left(\hat{p}_{i,j}^t > \frac{1}{2} + \epsilon_l + \hat{c}^t\right) \leq e^{-2t(\hat{c}^t)^2} = e^{-\log \frac{4t^2}{\delta}} = \frac{\delta}{4t^2}.$$

By the union bound, $Pr\left(\exists t \text{ s.t. } \hat{p}_{i,j}^t > \frac{1}{2} + \frac{\epsilon_l + \epsilon_u}{2} + \hat{c}^t\right) \leq \sum_t \frac{\delta}{4t^2} \leq \frac{\delta}{2}$.

Similarly, by Hoeffding's inequality,

$$Pr\left(\hat{p}_{i,j}^m > \frac{1}{2} + \frac{\epsilon_l + \epsilon_u}{2}\right) \leq e^{-2m((\epsilon_u - \epsilon_l)/2)^2} = e^{-\log \frac{2}{\delta}} = \frac{\delta}{2}. \qquad \square$$

We now show that under the second hypothesis namely $\tilde{p}_{i,j} \geq \epsilon_u$, w.p.$\geq 1 - \delta$, COMPARE$(i, j, \epsilon_l, \epsilon_u, \delta)$ returns 2.

**Lemma 12.** *For $\epsilon_u > \epsilon_l \geq 0$, if $\tilde{p}_{i,j} \geq \epsilon_u$, then w.p.$\geq 1 - \delta$, COMPARE$(i, j, \epsilon_l, \epsilon_u, \delta)$ outputs 2.*

*Proof.* Let $\hat{p}_{i,j}^t$ and $\hat{c}^t$ denote $\hat{p}_{i,j}$ and $\hat{c}$ respectively after $t$ comparisons between $i$ and $j$ during COMPARE$(i, j, \epsilon_l, \epsilon_u, \delta)$. COMPARE$(i, j, \epsilon_l, \epsilon_u, \delta)$ outputs 1 only if $\hat{p}_{i,j}^t < \frac{1}{2} + \frac{\epsilon_l + \epsilon_u}{2} - \hat{c}^t$ for any $t < m = \frac{2}{(\epsilon_u - \epsilon_l)^2} \log \frac{2}{\delta}$ or if $\hat{p}_{i,j}^m \leq \frac{1}{2} + \frac{\epsilon_l + \epsilon_u}{2}$. We bound the probability of either of these events by $\frac{\delta}{2}$ and the result follows from the union bound.

By Hoeffding's inequality,

$$Pr\left(\hat{p}_{i,j}^t < \frac{1}{2} + \frac{\epsilon_l + \epsilon_u}{2} - \hat{c}^t\right) \leq Pr\left(\hat{p}_{i,j}^t < \frac{1}{2} + \epsilon_u - \hat{c}^t\right) \leq e^{-2t(\hat{c}^t)^2} = e^{-\log \frac{4t^2}{\delta}} = \frac{\delta}{4t^2}.$$

By the union bound, $Pr\left(\exists t \text{ s.t. } \hat{p}_{i,j}^t < \frac{1}{2} + \frac{\epsilon_l + \epsilon_u}{2} - \hat{c}^t\right) \leq \sum_t \frac{\delta}{4t^2} \leq \frac{\delta}{2}$.

Similarly, by Hoeffding's inequality,

$$Pr\left(\hat{p}_{i,j}^m \leq \frac{1}{2} + \frac{\epsilon_l + \epsilon_u}{2}\right) \leq e^{-2m((\epsilon_u - \epsilon_l)/2)^2} = e^{-\log \frac{2}{\delta}} = \frac{\delta}{2}. \qquad \square$$

Thus proof of Lemma 1 follows from Lemmas 10, 11 and 12.

### A.3 Proof of Theorem 2

*Proof.* We first bound the total number of comparisons. Before each call of COMPARE (step 7), SEQ-ELIMINATE eliminates an element in step 6, hence COMPARE is called for exactly $n - 1$ times. Further observe that COMPARE$(i, j, 0, \epsilon, \delta/n)$ always uses less than $\frac{2}{\epsilon^2} \log \frac{2n}{\delta}$ comparisons. Hence the total comparisons used by SEQ-ELIMINATE$(S, \epsilon, \delta)$ is

$$\leq \sum_{k=1}^{n-1} \frac{2}{\epsilon^2} \log \frac{2n}{\delta} = \mathcal{O}\left(\frac{n}{\epsilon^2} \log \frac{n}{\delta}\right).$$

We now show that w.p.$\geq 1 - \delta$, SEQ-ELIMINATE$(S, \epsilon, \delta)$ outputs an $\epsilon$-maximum. Let $r^t, c^t$ denote the running and competing elements respectively before $t^{th}$ run of COMPARE. Then by Lemmas 11 and 12, for any $t$, w.p. $\geq 1 - \frac{\delta}{n}$,

$$\tilde{p}_{r^{t+1}, r^t} \geq 0, \tag{1}$$

$$\tilde{p}_{r^{t+1}, c^t} > -\epsilon. \tag{2}$$

Further, by the union bound the probability that Equations 1 and 2 do not hold for some $1 \leq t \leq n$ is $\leq \delta$. Now let $b^*$ be the absolute maximum element i.e., $\tilde{p}_{b^*, e} \geq 0$ $\forall e \in S$. Then, either $b^*$ is set as the running element before the first run of COMPARE i.e., $r^1 = b^*$ or $b^*$ is the competing element at the $t^{th}$ run of COMPARE for some $1 \leq t \leq n$ i.e., $c^t = b^*$. We show that in both cases, the output is an $\epsilon$-maximum.

If $r^1 = b^*$, then by equation 1, future running elements are either $b^*$ or better than $b^*$. Since $b^*$ is the absolute maximum, future running elements must be $b^*$ and hence $b^*$ is the output.

If for some $t$, $c^t = b^*$, then by equation 2, $\tilde{p}_{r^{t+1}, b^*} > -\epsilon$. Further, by equation 1, $\tilde{p}_{r^l, r^{t+1}} \geq 0$ $\forall l \geq t + 1$. Hence by strong stochastic transitivity, $\tilde{p}_{r^n, b^*} > -\epsilon$. Again, by strong stochastic transitivity, $\tilde{p}_{r^n, e} > -\epsilon$ $\forall e \in S$. Hence, the output is $\epsilon$-maximum. $\qquad \square$

### A.4 PICK-ANCHOR algorithm

---
**Algorithm 5** PICK-ANCHOR
---
1: **inputs**
2:    Set $S$ of size $n$, size $n'$, bias $\epsilon$, confidence $\delta$.
3: Form a set $Q$ by selecting $\min\left(\frac{n}{n'} \log \frac{2}{\delta}, n\right)$ random elements from $S$ without replacement.
4: **return** SEQ-ELIMINATE$\left(Q, \epsilon, \frac{\delta}{2}\right)$

---

## A.5 Proof of Lemma 3

*Proof.* We first bound the number of comparisons used by PICK-ANCHOR$(S, n', \epsilon, \delta)$. Since $|Q| \leq \frac{n}{n'} \log \frac{2}{\delta} = \mathcal{O}(\frac{n}{n'} \log \frac{1}{\delta})$, Theorem 2 implies that the number of comparisons used by PICK-ANCHOR is

$$= \frac{2|Q|}{\epsilon^2} \log \frac{2|Q|}{\delta} = \mathcal{O}\left( \frac{\frac{n}{n'} \log \frac{1}{\delta}}{\epsilon^2} \log \frac{\frac{n}{n'} \log \frac{1}{\delta}}{\delta} \right)$$

$$= \mathcal{O}\left( \frac{n}{n'\epsilon^2} \log \frac{1}{\delta} \left( \log \frac{n}{n'\delta} + \log \log \frac{1}{\delta} \right) \right)$$

$$= \mathcal{O}\left( \frac{n}{n'\epsilon^2} \log \frac{1}{\delta} \log \frac{n}{n'\delta} \right).$$

We show that w.p.$\geq 1 - \delta$, the output element is an $(\epsilon, n')$-good anchor element. We first show that $Q$ contains atleast one of the top $n'$ elements. We then show that the output element defeats one of the top $n'$ elements with probability $\geq \frac{1}{2} - \epsilon$. Hence by strong stochastic transitivity, the output element defeats every element outside the top $n'$ elements with probability $\geq \frac{1}{2} - \epsilon$.

The probability that $Q$ does not contain an element in top $n'$ elements is $\leq \left( 1 - \frac{n'}{n} \right)^{\frac{n}{n'} \log \frac{2}{\delta}} \leq \frac{\delta}{2}$. Note that the above statement is true even when size of $Q$ is $n$. Let the best element in $Q$ be denoted as $q^*$. By Theorem 2, w.p.$\geq 1 - \delta/2$, the output element $o$ of SEQ-ELIMINATE$(Q, \epsilon, \frac{\delta}{2})$ is an $\epsilon$-maximum of $Q$. Hence w.p. $\geq 1 - \delta/2$, $\tilde{p}_{q^*,o} \leq \epsilon$ and therefore by strong stochastic transitivity, for element $e$ worse than $q^*$, $\tilde{p}_{e,o} \leq \tilde{p}_{q^*,o} \leq \epsilon$. Since, w.p.$\geq 1 - \delta/2$, the number of elements that are better than $q^*$ is less than $n'$, by the union bound, w.p.$\geq 1 - \delta$, $o$ is an $(\epsilon, n')$-good anchor element. $\square$

## A.6 Proof of Lemma 5

We prove the Lemma by dividing it into three parts. We first show that an element $e$ that is $\epsilon_u$ better than anchor $a$ i.e., $\tilde{p}_{e,a} \geq \epsilon_u$, is part of PRUNE$(S, a, n', \epsilon_l, \epsilon_u)$ w.p.$\geq 1 - \delta/2$.

**Lemma 13.** *If $\tilde{p}_{e,a} \geq \epsilon_u$, then w.p.$\geq 1 - \delta/2$, the output set of PRUNE$(S, a, n', \epsilon_l, \epsilon_u, \delta)$ contains $e$.*

*Proof.* $e$ is not part of output set only if $e \in Q_t$ for some $t$. $Q_t$ will contain $e$ only if $S_t$ contains $e$ and COMPARE$(e, a, \epsilon_l, \epsilon_u, \frac{\delta}{2^{t+1}})$ returns 1. By Lemma 12, since $\tilde{p}_{e,a} \geq \epsilon_u$, probability that COMPARE$(e, a, \epsilon_l, \epsilon_u, \frac{\delta}{2^{t+1}})$ returns 1 is $\leq \frac{\delta}{2^{t+1}}$. Hence the probability that $Q_t$ contains $e$ is $\leq \frac{\delta}{2^{t+1}}$ and therefore by the union bound the probability that output set does not contain $e$ is $\leq \sum_{t=1}^{\infty} \frac{\delta}{2^{t+1}} \leq \frac{\delta}{2}$. $\square$

For $\frac{1}{n} \leq \delta \leq \frac{n'}{n}$, and if $a$ is a good anchor element, we show that first round of pruning itself will reduce the set size to $2n'$ and hence bound the number of comparisons used by PRUNE.

**Lemma 14.** *If $n' \geq 8 \log^2 n$, $\frac{1}{n} \leq \delta \leq \frac{n'}{n}$ and $a$ is an $(\epsilon_l, n')$-good anchor element then w.p.$\geq 1 - \delta/2$, PRUNE$(S, a, n', \epsilon_l, \epsilon_u, \delta)$ uses $\mathcal{O}\left( \frac{n}{(\epsilon_u - \epsilon_l)^2} \log \frac{1}{\delta} \right)$ comparisons and outputs a set of size at most $2n'$.*

*Proof.* If $n' \geq \frac{n}{2}$, the lemma is trivial. So let $n' < \frac{n}{2}$. Let the elements that defeat $a$ with probability $\geq \frac{1}{2} + \epsilon_l$ i.e., elements in set $\{e|\tilde{p}_{e,a} \geq \epsilon_l\}$ be called good elements and the remaining elements be bad elements. Note that the number of bad elements in $S_1$ is $\geq n - n'$. We show that the number of bad elements in $S_2$ is $\leq n'$. An element $e$ is part of $S_2$ only if COMPARE$(e, a, \epsilon_l, \epsilon_u, \delta/4)$ returns 2. By Lemma 11, each bad element in $S_1$ appears in $S_2$ w.p.$\leq \delta/4$. Therefore by the Chernoff bound, the probability that there are more than $n'$ bad elements in $S_2$ is

$$\leq e^{-(n-n')D\left(\frac{n'}{n-n'} \| \delta/4\right)} \leq e^{-\frac{n}{2} D\left(\frac{n'}{n} \| \delta/4\right)} \leq e^{-\frac{n}{2} \frac{n'}{2n}} = e^{-n'/4} \leq \frac{1}{n^2} \leq \frac{\delta}{2}.$$

Since the number of good elements in $S_1$ is $\leq n'$, their size in $S_2$ is also $\leq n'$. Hence w.p.$\geq 1 - \delta/2$, $|S_2| \leq 2n'$ and therefore PRUNE stops after first iteration. Noting that PRUNE ran only for one iteration $t = 1$, COMPARE$(e, a, \epsilon_l, \epsilon_u, \delta/4)$ uses $\mathcal{O}\left( \frac{1}{(\epsilon_u - \epsilon_l)^2} \log \frac{1}{\delta} \right)$ comparisons. $\square$

We now bound the number of comparisons used by PRUNE for higher values of $\delta$ by showing that after each round, the number of elements reduces roughly by a factor of $\delta$.

**Lemma 15.** *If $n' > \sqrt{6n \log n}$, $\delta \geq \frac{n'}{n}$ and $a$ is an $(\epsilon_l, n')$-good anchor element, then w.p.$\geq 1 - \frac{\delta}{2}$, PRUNE$(S, a, n', \epsilon_l, \epsilon_u, \delta)$ uses $\mathcal{O}\left( \frac{n}{(\epsilon_u - \epsilon_l)^2} \log \frac{1}{\delta} \right)$ comparisons and outputs a set of size less than $2n'$.*

*Proof.* As before let the elements that defeat $a$ with probability $\geq \frac{1}{2} + \epsilon_l$ i.e., elements in set $\{e | \tilde{p}_{e,a} \geq \epsilon_l\}$ be called good elements and the remaining elements be bad elements. The number of good elements in $S_1$ is $\leq n'$ and number of bad elements in $S_1$ is $\geq n - n'$. We first show that in each iteration the number of bad elements decreases by atleast a factor of $\delta$ until it falls below $n'$. We then bound the number of rounds it takes for number of bad elements to fall below $n'$. Using this bound on number of rounds, we separately bound the number of comparisons used over bad and good elements.

Note that for every bad element $e$, COMPARE$(e, a, \epsilon_l, \epsilon_u, \delta')$ outputs 2 with probability $\leq \delta' \leq \delta/4$. Hence, if at the beginning of the round, the number of bad elements is more than $n'$, the probability that number of bad elements does not reduce by at least a factor of $\delta$ is

$$\leq e^{-n' D(\delta \| \delta/4)} \leq e^{-n'\delta/2} \leq e^{-\frac{(n')^2}{2n}} \leq \frac{1}{n^3}$$

where the last inequality follows from $n' \geq \sqrt{6n \log n}$.

Now if the number of bad elements reduces by $\delta$ after each round, then the number of bad elements falls below $n'$ in $t = 2 \log_{\frac{1}{\delta}} \frac{n}{n'} \leq n$ rounds. Thus by the union bound, w.p.$\geq 1 - \frac{1}{n^2}$, the number of bad elements reduces by $\delta$ until the size becomes less than $n'$. Henceforth we assume this and bound the number of comparisons used.

We first bound the number of comparisons taken by PRUNE over bad elements. Number of bad elements in $S_t$ is $\leq n\delta^{t-1}$. Since COMPARE$(e, a, \epsilon_l, \epsilon_u, \delta')$ uses $\frac{2}{(\epsilon_u - \epsilon_l)^2} \log \frac{1}{\delta'}$, the number of comparisons used by PRUNE over bad elements is

$$\leq \sum_{t=1}^{2\log_{1/\delta} \frac{n}{n'}} \frac{2n\delta^{t-1}}{(\epsilon_u - \epsilon_l)^2} \log \frac{2^{t+1}}{\delta}$$

$$\leq \frac{2n}{(\epsilon_u - \epsilon_l)^2} \sum_{t=1}^{2\log_{1/\delta} \frac{n}{n'}} \left( \delta^{t-1} \log \frac{1}{\delta} + (t+1)(\delta)^{t-1} \log 2 \right)$$

$$= \mathcal{O}\left( \frac{n}{(\epsilon_u - \epsilon_l)^2} \log \frac{1}{\delta} \right).$$

The last equality follows from the fact that if $\delta \leq 1/2$ (if $\delta > 1/2$, we can choose $\delta = 1/2$) then $\sum_t \delta^{t-1}$ and $\sum_t (t+1)\delta^{t-1}$ are bounded.

Now we bound the number of comparisons used by PRUNE over good elements. The number of comparisons used by PRUNE over good elements is

$$\leq \sum_{t=1}^{2\log_{1/\delta} \frac{n}{n'}} \frac{n'}{(\epsilon_u - \epsilon_l)^2} \log \frac{2^{t+1}}{\delta}$$

$$\leq \frac{n'}{(\epsilon_u - \epsilon_l)^2} \sum_{t=1}^{2\log_{1/\delta} \frac{n}{n'}} \left( \log \frac{1}{\delta} + (t+1) \log 2 \right)$$

$$\leq \frac{n'}{(\epsilon_u - \epsilon_l)^2} \left( \left( 2 \log_{1/\delta} \frac{n}{n'} \right) \log \frac{1}{\delta} + \left( 2 \log_{1/\delta} \frac{n}{n'} \right)^2 \right)$$

$$= \mathcal{O}\left( \frac{n}{(\epsilon_u - \epsilon_l)^2} \log \frac{1}{\delta} \right). \qquad \square$$

Proof of Lemma 5 follows from Lemmas 13, 14 and 15.

## A.7 Proof of Theorem 6

We prove the theorem by breaking it into parts. We first show that if anchor element $a$, the output of PICK-ANCHOR is a $2\epsilon/3$-maximum then w.p.$\geq 1 - \delta/4$, OPT-MAXIMIZE outputs $a$.

**Lemma 16.** *If $a$, the output of step 6 in* OPT-MAXIMIZE$(S, \epsilon, \delta)$ *is a $\frac{2\epsilon}{3}$-maximum of $S$, then w.p.$\geq 1 - \frac{\delta}{4}$,* OPT-MAXIMIZE$(S, \epsilon, \delta)$ *outputs $a$.*

*Proof.* $a$ is not returned only if COMPARE in step 9 of OPT-MAXIMIZE returns 2. Since $a$ is $\frac{2\epsilon}{3}$-maximum of $S$, $\tilde{p}_{e,a} \leq \frac{2\epsilon}{3}$, $\forall e \in S$. Then by Lemma 11, the probability that a single call of COMPARE$(e, a, 2\epsilon/3, \epsilon, \frac{\delta}{4n})$ returns 2 is $\leq \frac{\delta}{4n}$. Hence by the union bound, the probability that COMPARE returns 1 for all calls in step 9 of OPT-MAXIMIZE is $\geq 1 - \delta/4$. Therefore the Lemma follows. $\square$

We now bound the number of comparisons used by OPT-MAXIMIZE in steps 1-6 and also prove some properties of PRUNE 's output set and anchor element.

**Lemma 17.** *For $\delta \geq \frac{1}{n}$, w.p.$\geq 1 - \delta/2$, steps 1-6 in OPT-MAXIMIZE$(S, \epsilon, \delta)$ uses $\mathcal{O}\left(\frac{n}{\epsilon^2} \log \frac{1}{\delta}\right)$ comparisons, outputs a set $S'$ of size at most $\sqrt{24n \log n}$ and either $a$ is a $2\epsilon/3$-maximum element or $S'$ contains the absolute maximum element.*

*Proof.* By Lemma 3, w.p.$\geq 1 - \delta/4$, PICK-ANCHOR$(S, \sqrt{6n \log n}, \frac{\epsilon}{3}, \frac{\delta}{4})$ uses $\mathcal{O}\left(\frac{\sqrt{n \log n}}{\epsilon^2} \log \frac{1}{\delta}\right)$ comparisons and outputs an $(\epsilon/3, \sqrt{6n \log n})$-good anchor element. From now we assume that $a$, the output of PICK-ANCHOR$(S, \sqrt{6n \log n}, \frac{\epsilon}{3}, \frac{\delta}{4})$ is an $(\epsilon/3, \sqrt{6n \log n})$-good achor element.

By Lemma 5, w.p.$\geq 1 - \delta/4$, PRUNE$(S, a, \sqrt{6n \log n}, \epsilon/3, 2\epsilon/3, \delta/4)$ uses $\mathcal{O}\left(\frac{n}{\epsilon^2} \log \frac{1}{\delta}\right)$ comparisons, outputs a set of size at most $\sqrt{24n \log n}$ and if $a$ is not an $2\epsilon/3$-maximum, then $S'$ contains the absolute maximum.

And the Lemma follows by using the union bound. $\square$

We now bound the number of comparisons used by OPT-MAXIMIZE during steps 8-13 assuming that either anchor element $a$ is $2\epsilon/3$-maximum or $S'$ contains the absolute maximum of $S$.

**Lemma 18.** *For $\delta \geq \frac{1}{n}$, if $a$, the output of step 6 and $S'$, the output of step 7 are such that either $a$ is $2\epsilon/3$-maximum of $S$ or $S'$ contains the absolute maximum element of $S$, then steps 8-13 of OPT-MAXIMIZE$(S, \epsilon, \delta)$ uses $\mathcal{O}\left(\frac{|S'|}{\epsilon^2} \log \frac{n}{\delta}\right)$ comparisons and w.p.$\geq 1 - \delta/2$, outputs an $\epsilon$-maximum.*

*Proof.* We first bound the number of comparisons. Each COMPARE$(e, a, 2\epsilon/3, \epsilon, \frac{\delta}{4n})$ uses $\mathcal{O}\left(\frac{1}{\epsilon^2} \log \frac{n}{\delta}\right)$ comparisons and hence over all elements of $S'$, COMPARE uses at most $\mathcal{O}\left(\frac{|S'|}{\epsilon^2} \log \frac{n}{\delta}\right)$ comparisons. Further SEQ-ELIMINATE$(S', \epsilon, \delta/4)$ uses $\mathcal{O}\left(\frac{|S'|}{\epsilon^2} \log \frac{n}{\delta}\right)$ comparisons by Theorem 2.

If $a$ is a $\frac{2\epsilon}{3}$-maximum, then the result follows by Lemma 16.

Let $a$ not be an $\frac{2\epsilon}{3}$-maximum. Then $S'$ contains the absolute maximum denoted here by $b^\star$. Notice that by strong stochastic transitivity, an $\epsilon$-maximum of $S'$ is an $\epsilon$-maximum of $S$ since $b^\star \in S'$. By Theorem 2, w.p.$\geq 1 - \frac{\delta}{4}$, SEQ-ELIMINATE$(S', \epsilon, \delta/4)$ outputs an $\epsilon$-maximum. Now if $\tilde{p}_{b^\star, a} > \epsilon$, then w.p.$\geq 1 - \frac{\delta}{4n}$, COMPARE$(b^\star, a, 2\epsilon/3, \epsilon, \frac{\delta}{4n})$ returns 2 and hence $a$ is not returned but SEQ-ELIMINATE$(S', \epsilon, \delta/4)$ is returned. If $\tilde{p}_{b^\star, a} \leq \epsilon$, then $a$ is an $\epsilon$-maximum and hence returning $a$ also results in an $\epsilon$-maximum output. Lemma then follows by the union bound. $\square$

Theorem 6 then follows from Theorem 2 and Lemmas 17 and 18.

# B  Ranking

## B.1  Proof sketch for Theorem 7

*Proof sketch.* Consider the model where $\tilde{p}_{a_1, a_n} = 1/2$, $\tilde{p}_{a_i, a_j} = (0 <)\mu (\ll 1/n^{10})$, when $i < j$ and $(i, j) \neq (n, 1)$. This model has an order: $a_1 > a_2 > \cdots > a_{n-1} > a_n$ i.e., $\tilde{p}_{a_i, a_j} > 0 \; \forall i < j$. Further this model satisfies strong stochastic transitivity since $\tilde{p}_{a_i, a_k} \geq \max(\tilde{p}_{a_i, a_j}, \tilde{p}_{a_j, a_k}) \; \forall i < j < k$.

We prove the Lemma by reducing the above model to the model where $\mu$ is replaced by 0. Note that new model does not satisfy strong stochastic transitivity but helps us in proving the Lemma.

Note that $\mu$ is so small that if we consider a model where we replace $\mu$ with 0, the comparisons behave essentially similarly. More formally, let model $M_\mu$ be the model we consider and $M_0$ be the model when $\mu$ is replaced with 0. Let $S$ denote a sequence of comparisons where each element of the sequence includes the elements compared and its outcome. Further, for each sequence $S$, let $P_\mu(S)$ and $P_0(S)$ denote the probability of sequence $S$ under models $M_\mu$ and $M_0$ respectively. Now consider a sequence $S$ of comparisons of length $\leq n^2/20$. Then

$$\frac{P_0(S)}{P_\mu(S)} \geq \left(\frac{1/2}{1/2 + \mu}\right)^{n^2/20} \geq e^{-n^2/(10n^{10})} \geq \frac{6}{7}$$

Thus the probability of any sequence of length $\leq \frac{n^2}{20}$ is approximately same under both models. Hence if there is an algorithm that uses $\frac{n^2}{20}$ comparisons and w.p.$\geq 7/8$ produces an $1/4$-ranking under $M_\mu$ model then applying same algorithm over $M_0$ model produces an $1/4$-ranking w.p.$\geq \frac{7}{8} \cdot \frac{6}{7} = \frac{3}{4}$.

We now show that there exists no algorithm that uses $\frac{n^2}{20}$ comparisons and w.p.$\geq \frac{3}{4}$ generates a 1/4-ranking under $M_0$, thus proving the Lemma. It is easy to see that any ordering outputted without querying the comparison between $a_1$ and $a_n$ is a 1/4-ranking w.p. exactly 1/2 since no order between $a_1$ and $a_n$ can be deduced. Since the pair $(a_1, a_n)$ is one random pair among $\binom{n}{2}$ pairs, the probability that the algorithm asks a comparison between this pair with $n^2/20$ comparisons is $< \frac{1}{2}$. So the probability that the output order contains $a_1$ and $a_n$ in the right order is $< \frac{1}{2} + \frac{1}{2} \cdot \frac{1}{2} = \frac{3}{4}$. $\qquad\square$

## B.2  Ranking Algorithm

We present STRONG-TRANSITIVITY-RANKING that uses $\mathcal{O}(\frac{n^2}{\epsilon^2} \log \frac{n}{\delta})$ comparisons and w.p.$\geq 1 - \delta$ outputs an $\epsilon$-ranking. STRONG-TRANSITIVITY-RANKING achieves this by approximating each $\tilde{p}_{i,j}$ with $\hat{p}_{i,j}$ to an additive error of $\frac{\epsilon}{2}$. We first argue that there is an element $e$ such that $\hat{p}_{e,j} \geq \frac{1}{2} - \epsilon/2 \; \forall j \in S$ and such an element is an $\epsilon$-maximum. Observe that if there is any element $e$ such that $\hat{p}_{e,j} \geq \frac{1}{2} - \epsilon/2 \; \forall j \in S$ then $p_{e,j} \geq \frac{1}{2} - \epsilon \; \forall j \in S$ and hence $e$ is an $\epsilon$-maximum of $S$. Further recall that for the absolute maximum $a^*$, $\tilde{p}_{a^*,j} \geq \frac{1}{2} \; \forall j \in S$ and hence $\hat{p}_{a^*,j} \geq \frac{1}{2} - \epsilon/2 \; \forall j \in S$. Therefore there will be at least one element $e$ s.t. $\hat{p}_{e,j} \geq \frac{1}{2} - \epsilon/2$ and such an element will be an $\epsilon$-maximum of $S$. We find one such element, delete it from $S$ and add it to the end of the ordered output set. We continue this process until we run out of elements in $S$. Since at every step we are adding an $\epsilon$-maximum of the remaining set, the ordered output set will be an $\epsilon$-ranking. We first present a subroutine ESTIMATE-PROBABILITY that compares two elements $a$ and $b$ for $\frac{1}{2\epsilon^2} \log \frac{2}{\delta}$ and w.p.$\geq 1 - \delta$ approximates $p(i,j)$ to an additive error of $\epsilon$.

---

**Algorithm 6** ESTIMATE-PROBABILITY

---

1: **inputs**
2:     element $i$, element $j$, bias $\epsilon$, confidence $\delta$.
3: Compare $i$ and $j$ for $\frac{1}{2\epsilon^2} \log \frac{2}{\delta}$ times.
4: **return** Fraction of times $i$ won

---

**Lemma 19.** ESTIMATE-PROBABILITY$(i, j, \epsilon, \delta)$ *uses* $\frac{1}{2\epsilon^2} \log \frac{2}{\delta}$ *comparisons and w.p.$\geq 1 - \delta$ approximates* $p_{i,j}$ *to an additive error of $\epsilon$.*

*Proof.* Proof follows from Hoeffding's inequality. $\qquad\square$

---

**Algorithm 7** STRONG-TRANSITIVITY-RANKING

---

1: **inputs**
2:     Set $S$, bias $\epsilon$, confidence $\delta$
3: **for** every pair $\{i, j\}$ such that $i, j \in S$ **do**
4:     $\hat{p}_{i,j} \leftarrow$ ESTIMATE-PROBABILITY$(i, j, \epsilon/2, \delta/n^2)$
5:     $\hat{p}_{j,i} \leftarrow 1 - p(i,j)$
6: **end for**
7: ordered set $T \leftarrow \varnothing$
8: **while** $|S| > 0$ **do**
9:     **if** $\exists e$ s.t. $\hat{p}_{e,f} \geq \frac{1}{2} - \epsilon \; \forall f \in S$ **then**
10:         Add $e$ at the end of $T$
11:         $S = S \smallsetminus \{e\}$
12:     **else**
13:         Add $S$ at the end of $T$
14:         **return** $T$
15:     **end if**
16: **end while**
17: **return** $T$

---

**Lemma 20.** STRONG-TRANSITIVITY-RANKING$(S, \epsilon, \delta)$ *uses* $\mathcal{O}(\frac{n^2}{\epsilon^2} \log \frac{n}{\delta})$ *comparisons and w.p.$\geq 1 - \delta$ returns an $\epsilon$-ranking.*

*Proof.* STRONG-TRANSITIVITY-RANKING calls ESTIMATE-PROBABILITY for $\mathcal{O}(n^2)$ times, once for each pair and each $EP(i, j, \epsilon/2, \delta/n^2)$ uses $\mathcal{O}(\frac{1}{\epsilon^2} \log \frac{n}{\delta})$ comparisons and hence bound on comparisons follow.

W.p.$\geq 1 - \delta/n^2$, ESTIMATE-PROBABILITY$(i, j, \epsilon/2, \delta/n^2)$ approximates $p_{i,j}$ with $\hat{p}_{i,j}$ such that $|p_{i,j} - \hat{p}_{i,j}| \leq \frac{\epsilon}{2}$. By the union bound, w.p.$\geq 1 - \delta$, $|p_{i,j} - \hat{p}_{i,j}| \leq \frac{\epsilon}{2}$ $\forall i, j, \in S$. From here we assume that $|p_{i,j} - \hat{p}_{i,j}| \leq \frac{\epsilon}{2}$ $\forall i, j, \in S$ and show that the output is an $\epsilon$-ranking. Let $S^t$ denote the set of remaining elements in $S$ after $t$ elements are removed from $S$. We first show that for $0 \leq t \leq n - 1$, there is one element $e$ such that $\hat{p}_{e,j} \geq \frac{1}{2} - \epsilon$ $\forall j \in S^t$ and such an element is an $\epsilon$-maximum of $S^t$. Observe that if there is an element $e$ such that $\hat{p}_{e,j} \geq \frac{1}{2} - \epsilon/2$ $\forall j \in S^t$ then $p_{e,j} \geq \frac{1}{2} - \epsilon$ $\forall j \in S^t$ and hence $e$ is an $\epsilon$-maximum of $S^t$. Further recall that for the absolute maximum $a^{t*}$ of $S^t$, $p_{a^{t*},j} \geq \frac{1}{2}$ $\forall j \in S^t$ and hence $\hat{p}_{a^{t*},j} \geq \frac{1}{2} - \epsilon/2$ $\forall j \in S^t$. Therefore there will be at least one element $e$ s.t. $\hat{p}_{e,j} \geq \frac{1}{2} - \epsilon/2$ and such an element will be an $\epsilon$-maximum of $S^t$. STRONG-TRANSITIVITY-RANKING deletes one such element from $S^t$ and adds it to the end of the ordered output set. Since for every $t$, STRONG-TRANSITIVITY-RANKING adds an $\epsilon$-maximum of $S^t$ to the output set, the Lemma follows. $\square$

## C  Borda Scores

### C.1  Ranking Algorithm for Borda Scores

---
**Algorithm 8** ESTIMATE-BORDA-SCORE
---
1: **inputs**
2:     set $S$, element $e$, bias $\epsilon$, confidence $\delta$.
3: **Initialize:** $w \leftarrow 0$, $\hat{s} \leftarrow \frac{1}{2}$, $m \leftarrow \frac{1}{2\epsilon^2} \log \frac{2}{\delta}$.
4: **for** $k = 1$ **to** $k = m$ **do**
5:     Compare $e$ with random element $\in S$
6:     **if** $e$ wins **then**
7:         $w \leftarrow w + 1$
8:     **end if**
9:     $\hat{s} = \frac{w}{k}$
10: **end for**
11: **return** $\hat{s}$

---

**Lemma 21.** ESTIMATE-BORDA-SCORE$(S, a, \epsilon, \delta)$ *uses* $\frac{1}{2\epsilon^2} \log \frac{2}{\delta}$ *comparisons and w.p.$\geq 1 - \delta$ approximates* $s(a)$ *to an additive error of $\epsilon$.*

*Proof.* Proof follows from properties of ESTIMATE-BORDA-SCORE and Hoeffding's inequlity. $\square$

---
**Algorithm 9** BORDA-RANKING
---
1: **inputs**
2:     set $S$, bias $\epsilon$, confidence $\delta$.
3: **Initialize:** $b_e \leftarrow \frac{1}{2}$ for all $e \in S$
4: **for** element $e$ in $S$ **do**
5:     $b_e \leftarrow$ ESTIMATE-BORDA-SCORE$(S, e, \frac{\epsilon}{2}, \frac{\delta}{n})$
6: **end for**
7: Rank $S$ according to $b_e$.
8: **return** $S$.

---

**Theorem 22.** BORDA-RANKING$(S, \epsilon, \delta)$ *uses* $\frac{2n}{\epsilon^2} \log \frac{2n}{\delta}$ *comparisons and w.p.$\geq 1 - \delta$ outputs an $\epsilon$-Borda ranking.*

*Proof.* BORDA-RANKING calls ESTIMATE-BORDA-SCORE for exactly $n$ times and each call of ESTIMATE-BORDA-SCORE$(S, e, \epsilon/2, \delta/n)$ uses $\frac{2}{\epsilon^2} \log \frac{2n}{\delta}$ comparisons and hence the bound on comparisons follows.

Note that w.p.$\geq 1 - \delta/n$, ESTIMATE-BORDA-SCORE$(S, e, \epsilon/2, \delta/n)$ approximates the Borda score of $e$ to an additive error of $\epsilon/2$. Let the approximate Borda score of element $e$ be $b_e$. By the union bound, w.p.$\geq 1 - \delta$, BORDA-RANKING$(S, \epsilon, \delta)$ approximates all Borda scores to an additive error of $\epsilon/2$. From here, we assume that $|b_e - s(e)| \leq \frac{\epsilon}{2}$ and show that ranking based on approximate Borda scores results in an $\epsilon$-Borda ranking.

If an element $e$ appears before element $f$ in the output ranking then $b_e \geq b_f$. Since $|b_e - s(e)| \leq \frac{\epsilon}{2}$ and $|b_f - s(f)| \leq \frac{\epsilon}{2}$, $s(e) - s(f) = (b_e - b_f) + (s(e) - b_e) + (b_f - s(f)) \leq \epsilon$. Hence the Lemma follows. $\square$

# D   Why Knockout Fails

We will show that KNOCKOUT proposed in [14] fails under SST model without stochastic triangle inequality constraint.

Consider the model where $\tilde{p}_{a_1,a_j} = \mu \ \forall j < n/2$, $\tilde{p}_{a_1,a_j} = \frac{1}{2} \ \forall j \geq n/2$ and $\tilde{p}_{a_i,a_j} = \mu \ \forall 1 < i < j$ for some $0 < \mu < \frac{1}{n^{10}}$ . Observe that this model satisfies SST but not stochastic triangle inequality. Under this model, $a_1$ is the absolute maximum and any element in the set $\{a_i | i < n/2\}$ is a $1/4$-maximum. We show that under this model, w.p.$\geq 1/16$, KNOCKOUT$(S, 1/4, 1/16)$ fails to find a $1/4$-maximum.

KNOCKOUT pairs elements randomly in each round and compares each pair for a certain number of times and the winners proceed to the next round until there is only one element left. Observe that in the first round $a_1$ can get paired with an element from set $\{a_i | 1 < i < n/2\}$ w.p.$\approx 1/2$ and if that happens $a_1$ can lose the tie w.p.$\approx 1/2$. Hence $a_1$ can get eliminated in the first round w.p.$\approx 1/4$. Once $a_1$ is eliminated, in the second round, the elements will be approximately half from the first half of the original set and half from the second half. Since these elements are almost incomparable (comparisons between any two elements is now approximately a Bernoulli random variable with parameter $1/2$), each element is almost equally likely to be the final output. Therefore w.p.$\approx 1/8$, the output can be an element from second half of the set and hence not a $1/4$-maximum.

# E   Additional Experiment

To show why PAC maximum algorithms could be preferred to absolute maximum algorithms, once again we compare SEQ-ELIMINATE, **KNOCKOUT** and **MallowsMPI** for comparison probability values close to $1/2$. In Figure 4, we consider the stochastic model, $p_{1,j} = 0.6 \ \forall j > 1$ and $p_{i,j} = 0.5 + \tilde{q} \ \forall 1 < i < j$ where $\tilde{q} \ll 0.05$ with $n = 15$. Again, we find a $0.05$-maximum with error probability $\delta = 0.1$. From Figure 4, we can observe that performance of **MallowsMPI** gets much worse as $\tilde{q}$ decreases whereas SEQ-ELIMINATE and **KNOCKOUT** do not get affected since they are PAC maxing algorithms. Also observe that SEQ-ELIMINATE performs much better than **KNOCKOUT**.

Figure 4: Comparison of Maximum Selection Algorithms for probability values close to $1/2$