[Reviews · NeurIPS 2017]

Reviewer 1



The paper considers a PAC setting with implicit feedback, namely the arms can be only compared in a pairwise manner. Four research questions are addressed in the paper: assuming strong stochastic transitivity, can the best item be found using O(n) samples and the ranking using O(n log n) samples. The same questions are addressed for Borda ranking for general preference matrix. The authors gave positive answers for all question except the second. The idea of the proposed algorithm is to distinguish low and high confidence cases (whether \delta < 1/n). The low confidence case is handled by using a sequential elimination strategy, whereas in the high confidence case, an anchor item is chosen which is among the top-k items, and then this is used to eliminate the worst half of the items. This algorithm is nice. The paper is well-written and easy to follow. The questions that are addressed are of importance. In fact, the negative result is somewhat surprising to me. I checked the supplementary material as well, it seems ok. And the supplementary material is also easy to follow. I have some minor comments: The following paper might be related: Kevin G. Jamieson and Sumeet Katariya and Atul Deshpande and Robert D. Nowak: Sparse Dueling Bandits, AISTAT, 2015. The paper shows that without assumoption on the preference matrix, the lower bound fo finding the Borda winner is \Omega( n^2). That is why it might be worth to cite this paper. The realted work is already comprehensive. Regarding the experiments: all algorithm tested in the paper are based on some kind of elimination strategy. The PL-PAC algorithm presented in Szorenyi et al. Online rank elicitation for Plackett-Luce: A dueling bandits approach., is an exception to some extend since it implements a a sorting-based sampling. It would be interesting to compare its performance to the elimination based algorithms. In the experiments of that paper, it had been found that the elimination strategies have quite high sample complexity comparing to the sorting-based sampling. But of course, that algorithm assumes BTL about the preference matrix which is a strong assumption, therefore its accuracy might drop if the model assumption is not true.

Reviewer 2



Thanks for the rebuttal, which was quite helpful and clarified most of the issues raised in the review. The authors study the problem of PAC maximum selection (maxing) and ranking of n elements via random pairwise comparisons. Problems of that kind have recently attracted attention in the ML literature, especially in the setting of dueling bandidts. In this paper, the only assumption made is strong stochastic transitivity of the pairwise winning probabilisties. Under this assumption, the authors show that maxing can be performed with linearly many comparisons while ranking has a quadratic sample complexity. Further, for the so-called Borda ranking, maximum selection and ranking can be performed with O(n) and O(n log n) comparisons, respectively. The paper addresses an interesting topic and is well written. That being said, I'm not convinced that is offers significant new results. It's true that I haven't seen the first two results in exactly this form before, but there are very similar ones, and besides, the results are not very surprising. Further, the argument for the result regarding the non-existence of a PAC ranking algorithm with O(n log n) comparisons in the SST setting is not totally clear. More importantly, the last two results are already known. Theorem 8 this known as Borda reduction, see e.g., Sparse Dueling Bandits, Generic exploration and k-armed voting bandits, Relative upper confidence bound for the k-armed duelling bandit problem. Likewise, Theorem 9 is already known, see e.g., Crowdsourcing for Participatory Democracies, Efficient elicitation of social choice functions. Consequently, the only result that remains, being both novel and relevant, is the PAC maxing algorithm and its analysis. Nevertheless, a study of its performance on real data is missing, as well as a comparison with other algorithms. Indeed, the experiments are not very convincing, especially because only a simple synthetic setting is considered, and no experiments for (Borda) ranking are included. SEQ-ELIMINATE is a subroutine of OPT-MAXIMIZE, so does it really makes sense to compare the two? It would be rather more meaningful to compare one of the two with other maxing algorithms in the main paper and move their comparison, if necessary, to the appendix. Lines 189-299: models with other parameters should be considered. In all maxing experiments, it should be tested whether the found element is the actual maxima. Minor issues: - 51: an \epsilon-ranking (and not \epsilon-rankings) - 100: comparisons (and not comparison) - 117-118: Please reconsider the statement: "the number of comparisons used by SEQ-ELIMINATE is optimal up to a constant factor when \delta <= 1/n". \delta <= 1/n => n <= 1/\delta => O(n/\epsilon^2 \log(n/\delta)) = O(n/\epsilon^2 \log(1/\delta^2)) != \Omega(n/\epsilon^2 \log(1/\delta)) - 117-118: Please reconsider the statement: "but can be \log n times the lower bound for constant \delta", since n/\epsilon^2 \log(1/\delta) \times \log(n) != n/\epsilon^2 \log(n/\delta). - 120-122: It is not clear how should such a reduction together with the usage of SEQ-ELIMINATE lead to an optimal sample complexity up to constants. PLease elaborate. - 130-131: "replacing or retaining r doesn’t affect the performance of SEQ-ELIMINATE". The performance in terms of what? And why there is no effect? - 165: same comment as for 117-118 - 167: in 120 it was stated that S' should have a size of O(n/\log n), and here It's O(\sqrt(n\log n)) - 165-169: The procedure described here (the reduction + usage of SEQ-ELIMINATE) is different from the one described in 120-122. And again, it's not clear how should such a procedure results in an order-wise optimal algorithm. - 204: Lemma 12 does not exist. - 207: Lemma 11 does not exist. - 210: at least (and not atleast) - 215: \delta is missing in the arguments list of PRUNE - 221-222: same comment as for 165 and 117-118 - 238-239: The sample complexity is not apparent in both cases (distinguished in algorithm 2)